# Coverage and determinants of Intermittent Preventive Treatment in pregnancy (IPTp) in Cameroon, Guinea, Mali, and Nigeria

**M. D. Nahid Hassan Nishan**[1]*, **Khadiza Akter**[2]

**1** Department of Public Health, North South University, Dhaka, Bangladesh, **2** Department of Nursing, International University of Business Agriculture and Technology, Dhaka, Bangladesh

* nissan0808@yahoo.com

## Abstract

### Introduction

Malaria poses a serious issue for pregnant women in African regions. It is crucial to comprehend the various factors that impact receiving IPTp during ANC checkups. These are important for the well-being of both pregnant mothers and their unborn children. Therefore, this study aims to investigate the determinants of IPTp coverage among women in Cameroon, Guinea, Mali, and Nigeria.

### Methodology

This cross-sectional study utilized secondary data from the Malaria Indicator Surveys (MIS) across Cameroon, Guinea, Mali, and Nigeria, focusing on women who received IPTp during pregnancy with IPTp categorized dichotomously as "Yes" for ANC visits and "No" for other visits. Chi-squared tests were used to assess associations, and binary logistic regression was conducted to calculate adjusted odds ratios, confidence intervals, and p-values. Results were summarized in tables.

### Results

We found IPTp coverage during ANC visits was highest in Cameroon (98.6%), followed by Guinea (97.7%), Mali (97.1%), and lowest in Nigeria (95.5%). In Guinea, rural women were less likely to receive IPTp than urban women (AOR: 0.16, 95% CI: 0.07–0.41, p<0.001). In Mali, women who received 3 or more doses were less likely to receive IPTp at ANC (AOR: 0.48, p<0.01). In Nigeria, personal transport increased IPTp uptake (AOR: 1.88, p<0.01). In Cameroon, malaria prevention messages improved IPTp coverage (AOR: 3.12, p<0.05).

### Conclusions

This study highlights significant disparities in IPTp uptake, with rural Mali and Guinea facing lower coverage. In Nigeria, personal transport improved IPTp uptake. Targeted interventions are needed to improve ANC services and ensure equitable IPTp access across the study regions.

**Data Availability Statement:** Data is available Publicly from Demographic and Health Surveys (DHS) program. The official repository of the

publicly access dataset URL: https://dhsprogram.com/data/available-datasets.cfm Also, here's the direct dataset URL: Cameroon: https://dhsprogram.com/methodology/survey/survey-display-563.cfm Nigeria: https://dhsprogram.com/methodology/survey/survey-display-576.cfm Guinea: https://dhsprogram.com/methodology/survey/survey-display-571.cfm Mali: https://dhsprogram.com/methodology/survey/survey-display-574.cfm.

**Funding:** The author(s) received no specific funding for this work.

**Competing interests:** The authors have declared that no competing interests exist.

**Abbreviations: ANC**, Antenatal Care; **AOR**, Adjusted Odds Ratio; **CI**, Confidence Interval; **DHS**, Demographic and Health Survey; **IPTp**, Intermittent preventive treatment in pregnancy; **MIS**, Malaria Indicator Surveys; **WHO**, World Health Organization.

## Introduction

Malaria is a vector-borne disease mainly caused by an infectious parasite called *Plasmodium falciparum* [1]. Globally, malaria remains a significant public health concern due to its high mortality rates. In 2022, an estimated 608,000 deaths occurred as a result of malaria, with the vast majority—over 94%—reported from the African region [2]. The region continues to face the brunt of the disease, with pregnant women being particularly vulnerable [3]. It is estimated that up to 9.5 million pregnant women in Sub-Saharan Africa, including West and Central Africa, may acquire malaria annually, highlighting the critical need for targeted interventions [4, 5].

To combat this, following updated WHO guidelines, various malaria prevention strategies have been introduced, including interventions like Intermittent Preventive Treatment in pregnancy (IPTp) [6]. IPTp has been widely implemented as a key strategy and effective IPTp coverage is crucial in reducing the malaria burden among pregnant women in high-risk regions, where malaria coverage remains alarmingly high [7]. approximately 11 million pregnancies are exposed to malaria each year in Sub-Saharan Africa, which includes West and Central Africa [8]. In a Meta-analysis, it was found that several morbid disease states and outcomes which include anemia, low birthweight, preterm birth, and stillbirth, may significantly affect malarial patients during pregnancy [9]. Despite this adverse effect, malaria rarely shows signs and symptoms during pregnancy, which makes it difficult to diagnose and provide comprehensive treatment [10]. That's why pregnant women need to be under the supervision of special care in those malarial endemic regions [4].

Nigeria, for example, bears the highest malaria burden globally, with an estimated 68 million cases and 194,000 deaths in 2021 alone. The transmission risk is persistent throughout the year placing pregnant women at constant risk [11]. Moreover, In Cameroon, malaria is responsible for 48% of hospital admissions, contributes to 30% of all morbidity, and accounts for 67% of childhood mortality annually. The entire population, exceeding 22 million people, remains at risk of malaria infection each year [12, 13]. These figures underline the urgency of improving IPTp coverage to mitigate the disease's impact on maternal and child health.

While IPTp has been introduced across these countries, challenges persist in ensuring adequate coverage, particularly during antenatal care (ANC) visits [14]. Studies have shown that consistent ANC visits are crucial for ensuring that pregnant women receive the full regimen of IPTp doses [15, 16]. However, gaps in ANC attendance often result in incomplete dosing, increasing the risk of malaria during pregnancy [17–19].

While previous studies have addressed IPTp coverage, there remains a need to corroborate these findings and further explore the specific factors influencing IPTp uptake in countries like Cameroon, Guinea, Mali, and Nigeria. Therefore, this study aims to build on existing research by utilizing a distinct analytical approach to investigate these factors within the context of West and Central Africa. The novelty of this study lies in its country-specific focus, offering valuable insights into the determinants of IPTp coverage and supporting more tailored and effective interventions to improve antimalarial medication coverage for pregnant women through ANC visits in these high-burden areas.

## Materials and methods

### Study description

According to Wikipedia, there present 16 countries in West Africa and 9 countries in Central Africa [20, 21]. For this study, secondary datasets from the Demographic and Health Survey (DHS) were utilized, specifically focusing on the Malaria Indicator Survey (MIS) datasets.

Among these 25 countries, only eight MIS datasets from 2021 onwards were available. A systematic literature review was conducted to identify the key dependent and independent variables required for the analysis. After assessing these variables across the available datasets, if any necessary variables were missing, that dataset was excluded from the study. Based on the completeness and availability of up-to-date data, four countries were deemed suitable for comprehensive analysis. These countries include: Central Africa: Cameroon 2022 [22], West Africa: Guinea 2021 [23], Mali 2021 [24], and Nigeria 2021 [25].

## Study design & participants

This study used a cross-sectional design, utilizing data from the MIS conducted by the DHS Program in four countries: Cameroon, Guinea, Mali, and Nigeria. Cross-sectional studies collect data at a single point in time, making them suitable for analyzing the coverage of specific interventions or behaviors in a population. The MIS surveys collect data from all women of reproductive age, and for this study, specifically data on women who had received Intermittent Preventive Treatment in pregnancy (IPTp) during their pregnancy from the most recent MIS datasets were selected, ensuring a highly relevant and focused participant group for analysis. The DHS Program employs a two-stage sampling process. In the first stage, clusters, or Enumeration Areas (EAs), are randomly selected from a comprehensive list of geographical areas. In the second stage, households within each selected cluster are randomly chosen to participate in the survey [26].

IPTp coverage in the MIS datasets was assessed among women aged 15–49 who had a live birth or stillbirth in the two years preceding the survey. These women were asked whether they had received any doses of SP/Fansidar during their most recent pregnancy. After removing observations with missing values using listwise deletion, four cleaned datasets were finalized, containing only complete data across all key variables. This ensured the integrity of the dataset and that all participants had the necessary information for analysis. The final weighted sample sizes were: Cameroon (1,912), Guinea (1,906), Mali (3,935), and Nigeria (3,367), giving a total of 11,120 participants. To ensure the survey results accurately reflect the wider population, the DHS Program strongly recommends the use of weighting techniques. Weighting is applied to adjust for any over- or under-representation of particular groups in the sample, accounting for the complex survey design, including stratification and clustering. This process ensures that the findings are truly representative of the population from which the sample was drawn [27]. Following DHS guidelines, the appropriate weighting, stratification, and primary sampling units were declared before conducting the final analysis, improving both the precision and the representativeness of the results.

## Definition of study variables

The outcome variable in this study focused on whether pregnant women received IPTp during their ANC visits. In the MIS, data on IPTp receipt during ANC visits was collected by asking women who had a live birth or stillbirth in the two years preceding the survey whether they received any doses of SP/Fansidar during their pregnancy, specifically noting the source of the antimalarial treatment [28]. The MIS recorded this information under the variable "source of antimalarial during pregnancy," distinguishing between ANC visits and other facility visits or other sources such as private clinics or pharmacies. Data were collected based on women's responses recorded from health cards or, when health cards were unavailable, through recall during their ANC visits. For this study, women who received IPTp during their ANC visits were categorized as "Yes," while those who received IPTp from non-ANC sources or did not receive it at all were categorized as "No."

The independent variables were selected through a systematic review of existing literature on factors influencing IPTp coverage, followed by exploratory data analysis. Women's ages were divided into three groups which include Early reproductive age (<25 years), Mid-reproductive age (25–34 years), and Late reproductive age (>35 years). Residence variables were categorized as Rural and Urban. In the wealth index, the poorest and poorer were recoded into poor, keeping the Middle category intact and the richer and richest into the rich category due to their insufficient sample size. Education level was categorized as No education, Primary, Secondary, and Higher. Religious affiliation was recoded to binary classification Muslim, and other categories due to a lower number of observations.

The number of IPTp doses received was categorized into two groups: 0–2 doses and ≥3 doses, based on the number of times women took SP/Fansidar during pregnancy. The access to personal transport variable was categorized as Yes or No based on a few questions in which the respondents were asked whether they have bicycles, motorcycles/scooters, and trucks/cars in their household. Have any malaria prevention knowledge variable categorized as Yes or No based on these questions: some ways to prevent malaria are sleeping under mosquito nets, sleeping under insecticide-treated mosquito nets, using mosquito repellent, taking preventive medications, spraying the house with insecticide, and filling in stagnant waters. Similarly, the dummy variable of heard of any malaria message in media was also categorized as Yes or No, based on these questions: the sources of malaria messages that people heard or saw were radio, television, poster/billboard, newspaper/magazine, leaflet/brochure, social media, and SMS.

## Ethical approval

This study utilized secondary data from the DHS program, which follows strict ethical guidelines. The DHS program ensures that all necessary ethical approvals are obtained before data collection. Informed consent was obtained from all participants by the DHS program, in collaboration with the respective country's Ministry of Health. The data used in this analysis is publicly available and anonymized, ensuring participant confidentiality and privacy. As the study involved secondary data analysis, no additional ethical approval was required for this research.

## Statistical analysis

All analyses were conducted using Stata version 17, with a p-value of less than 0.05 considered statistically significant. Descriptive statistics summarized the factors across the four countries. Associations between independent variables and the outcome (IPTp receipt during ANC visits) were assessed using chi-squared ($\chi^2$) tests. Variables showing $p \leq 0.2$ in any of the countries were included in the multivariate logistic regression model to reduce the risk of missing potential factors.

The logistic regression adjusted for potential confounders, including maternal age, residence, wealth index, education, religion, IPTp dose received, transportation access, malaria prevention knowledge, and media exposure to malaria messages. Multicollinearity was checked using a correlation matrix between all the independent variables, with no multicollinearity found. Model fit was assessed using the Hosmer-Lemeshow goodness-of-fit test, where groups were set to 5, and a p-value > 0.05 indicated a well-fitting model. The final model presented adjusted odds ratios (AOR) with 95% confidence intervals (CI), which were displayed in a table.

## Results

Across the four countries, IPTp coverage during ANC visits was as follows: Cameroon showed the highest coverage with 1,885 women (98.59%), followed by Guinea with 1,863 women

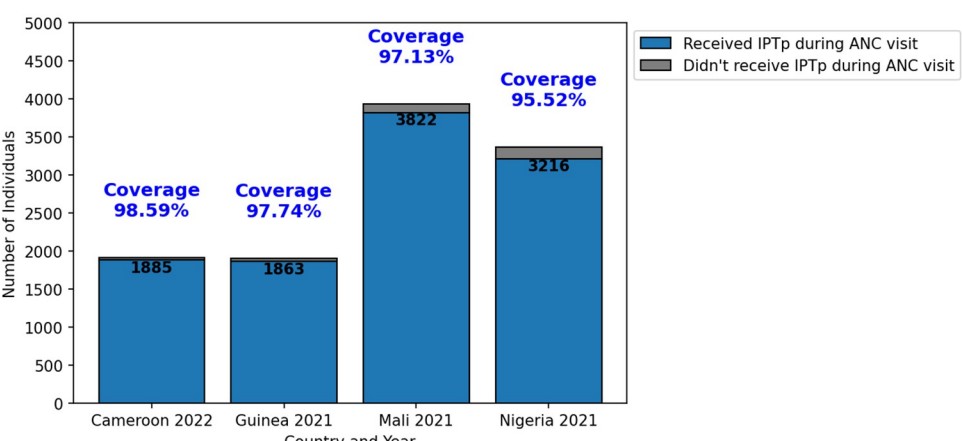

**Fig 1. Bar chart showing coverage of women receiving IPTp during antenatal care visits.**

(97.74%), and Mali with 3,822 women (97.13%). Nigeria exhibited the lowest coverage with 3,216 women (95.52%). These details can be seen in (**Fig 1**).

In **Table 1**, the descriptive analysis showed that Nigeria exhibits higher proportions of individuals in the mid-reproductive period (51.71%), whereas Mali had a lower representation in the late reproductive phase (19.83%). Guinea had the highest proportion of samples which were in the early reproductive phase (39.77%). Nigeria, in contrast, displayed a lower percentage (25.29%) in the early reproductive period, reflecting demographic diversity across the nations. In terms of residence, most people reside in rural areas, and Mali had the highest percentage of rural residences (78.04%), In terms of central African country, Cameroon demonstrates a higher proportion of people (48.94%) live in urban areas across the regions. Economic variations are depicted through the wealth index, with Guinea having a notable proportion of individuals classified as poor (42.10%), highlighting potential economic challenges in this country. Educational differences were prominent, notably in Mali, where a substantial portion (65.58%) lacks formal education. Conversely, Cameroon exhibits higher percentages of individuals with secondary (41.75%), and Nigeria shows higher education levels (13.73%), reflecting diverse educational landscapes. Distinct religious patterns are observed, with Mali having the highest percentage of Muslims (95.12%), and Cameroon showing the lowest (32.75%). The highest proportion of women receiving suboptimal IPTp doses (0–2 doses) was in Mali (55.36%), followed by Nigeria (47.62%). Cameroon (44.32%) and Guinea (41.99%) had lower proportions of women receiving 0–2 doses. In contrast, Guinea had the highest proportion of women receiving the full IPTp course (three or more doses) at 58.01%, followed by Cameroon (55.68%) and Nigeria (52.38%). Mali had the lowest proportion of women receiving three or more doses, at 44.64%. Insights from malaria-related contexts indicate a similar pattern, with the majority possessing malaria prevention knowledge whereas Cameroon possesses the highest percentage (89.59%). However, hearing any information in media regarding malarial messages is lower across all the regions, but the lowest was in Cameroon (80.76%). Additionally, having access to personal transport is notable, particularly in Mali, which has the highest access (83.34%).

## Demographic factors

When examining the IPTp coverage during ANC visits based on demographic factors across the four countries, as shown in **Table 2**, significant associations were observed in some cases.

**Table 1. Demographics tables of the respondents.**

|  | Central African Country | West African Country | | |
|---|---|---|---|---|
|  | Cameroon | Guinea | Mali | Nigeria |
| **Characteristics** | **N (%)** | **N (%)** | **N (%)** | **N (%)** |
| **Reproductive Age Group** |  |  |  |  |
| • Early Reproductive Period | 649 (33.94%) | 758 (39.77%) | 1,320 (33.56%) | 851 (25.29%) |
| • Mid-Reproductive Period | 939 (49.13%) | 843 (44.23%) | 1,834 (46.61%) | 1,742 (51.71%) |
| • Late Reproductive Period | 324 (16.94%) | 305 (16.00%) | 781 (19.83%) | 774 (23.00%) |
| **Residence** |  |  |  |  |
| • Urban | 936 (48.94%) | 532 (27.88%) | 864 (21.96%) | 1,150 (34.16%) |
| • Rural | 976 (51.06%) | 1,374 (72.12%) | 3,071 (78.04%) | 2,217 (65.84%) |
| **Wealth Index** |  |  |  |  |
| • Middle | 378 (19.78%) | 396 (20.78%) | 845 (21.48%) | 699 (20.76%) |
| • Poor | 778 (40.68%) | 802 (42.10%) | 1,451 (36.86%) | 1,131 (33.59%) |
| • Rich | 756 (39.53%) | 708 (37.12%) | 1,639 (41.65%) | 1,537 (45.64%) |
| **Education** |  |  |  |  |
| • No Education | 406 (21.23%) | 1,186 (62.23%) | 2,581 (65.58%) | 1,178 (34.98%) |
| • Primary | 506 (26.48%) | 401 (21.06%) | 591 (15.04%) | 563 (16.75%) |
| • Secondary | 798 (41.75%) | 264 (13.87%) | 696 (17.70%) | 1,163 (34.54%) |
| • Higher | 202 (10.54%) | 55 (2.84%) | 67 (1.68%) | 463 (13.73%) |
| **Religion** |  |  |  |  |
| • Muslim | 626 (32.75%) | 1,650 (86.56%) | 3,743 (95.12%) | 2,056 (61.07%) |
| • Others | 1286 (67.25%) | 256 (13.44%) | 192 (4.88%) | 1311 (38.93%) |
| **IPTp Doses Received** |  |  |  |  |
| • 0–2 | 848 (44.32%) | 800 (41.99%) | 2178 (55.36%) | 1603 (47.62%) |
| • ≥3 | 1064 (55.68%) | 1106 (58.01%) | 1757 (44.64%) | 1764 (52.38%) |
| **Have Any Malaria prevention knowledge** |  |  |  |  |
| • No | 199 (10.41%) | 413 (21.67%) | 424 (10.78%) | 636 (18.90%) |
| • Yes | 1713 (89.59%) | 1,493 (78.33%) | 3,511 (89.22%) | 2,731 (81.10%) |
| **Heard Any Malarial Massage in Media** |  |  |  |  |
| • No | 1544 (80.76%) | 1,256 (65.91%) | 2,546 (64.69%) | 2,411 (71.61%) |
| • Yes | 368 (19.24%) | 650 (34.09%) | 1,389 (35.31%) | 956 (28.39%) |
| **Has Access to Personal Transport** |  |  |  |  |
| • No | 1092 (57.14%) | 924 (48.52%) | 655 (16.66%) | 1,698 (50.43%) |
| • Yes | 820 (42.86%) | 982 (51.48%) | 3,280 (83.34%) | 1,669 (49.57%) |

In Cameroon and Nigeria, no significant associations were found in either the chi-squared or multivariate analyses of the residence variable. In Guinea, a higher percentage of rural women (64.03%) received IPTp during ANC visits compared to urban women (35.97%), but multivariate analysis revealed that rural women were significantly less likely to receive IPTp during ANC compared to urban women (AOR: 0.16, 95% CI: 0.07–0.41, p<0.001). In Mali, both the chi-squared ($X^2 = 80.00$, p<0.001) and multivariate analyses were significant. Urban women had higher IPTp coverage (57.20%) compared to rural women (42.80%) during ANC visits. In multivariate analysis, Rural women were significantly less likely to receive IPTp during ANC compared to urban women (OR: 0.14, 95% CI: 0.06–0.32, p<0.001).

Significant associations were found in IPTp coverage during ANC visits across education in Cameroon ($X^2 = 8.27$, p<0.05) and Nigeria ($X^2 = 12.01$, p<0.05), and across religion in Guinea ($X^2 = 5.01$, p<0.01), based on chi-squared tests. However, these associations did not remain

**Table 2. Association and factors influencing the receipt of IPTp during antenatal visits among the respondent.**

| Characteristics | Central African Country | | | | | | West African Country | | | | | |
|---|---|---|---|---|---|---|---|---|---|---|---|---|
| | Cameroon | | | Guinea | | | Mali | | | Nigeria | | |
| | N (%) | AOR | 95% CI | N (%) | AOR | 95% CI | N (%) | AOR | 95% CI | N (%) | AOR | 95% CI |
| **Reproductive Age Group** | $X^2 = 1.08$ | | | $X^2 = 0.94$ | | | $X^2 = 5.34$ | | | $X^2 = 0.44$ | | |
| • Early Reproductive Period | 7(25.42) | Ref | | 14(32.64) | Ref | | 28(24.85) | Ref | | 35(23.12) | Ref | |
| • Mid-Reproductive Period | 15(58.37) | 1.67 | 0.61–4.55 | 21(48.64) | 1.41 | 0.78–2.56 | 54(48.23) | 1.43 | 0.87–2.36 | 81(54.01) | 1.28 | 0.85–1.93 |
| • Late Reproductive Period | 5(16.21) | 1.27 | 0.37–4.32 | 7(18.72) | 1.64 | 0.63–4.27 | 31(26.91) | 1.78 | 0.92–3.45 | 34(22.87) | 1.22 | 0.72–2.08 |
| **Residence** | $X^2 = 0.42$ | | | $X^2 = 1.41$ | | | $X^2 = 80.00^{\Psi\Psi\Psi}$ | | | $X^2 = 2.41$ | | |
| • Urban | 15(55.14) | Ref | | 15(35.97) | Ref | | 65(57.20) | Ref | | 42(28.21) | Ref | |
| • Rural | 12(44.86) | 1.35 | 0.62–2.92 | 27(64.03) | 0.16 | 0.07–0.41*** | 48(42.80) | 0.14 | 0.06–0.32*** | 108(71.79) | 1.01 | 0.56–1.79 |
| **Wealth Index** | $X^2 = 2.72$ | | | $X^2 = 4.54$ | | | $X^2 = 24.27^{\Psi\Psi\Psi}$ | | | $X^2 = 8.41^{\Psi}$ | | |
| • Middle | 5(10.93) | Ref | | 4(9.09) | Ref | | 9(8.19) | Ref | | 36(23.62) | Ref | |
| • Poor | 10(35.29) | 1.82 | 0.40–8.34 | 23(54.94) | 4.31 | 1.34–13.79 | 32(28.10) | 1.99 | 1.04–3.81* | 63(42.21) | 1.08 | 0.61–1.88 |
| • Rich | 12(53.78) | 2.30 | 0.69–7.68 | 15(35.97) | 0.59 | 0.12–2.72 | 72(63.71) | 1.25 | 0.52–3.04 | 51(34.17) | 0.83 | 0.49–1.43 |
| **Education** | $X^2 = 8.27^{\Psi}$ | | | $X^2 = 2.46$ | | | $X^2 = 8.15$ | | | $X^2 = 12.01^{\Psi}$ | | |
| • No Education | 5(18.51) | Ref | | 28(67.11) | Ref | | 62(54.53) | Ref | | 67(44.48) | Ref | |
| • Primary | 6(22.22) | 0.67 | 0.11–3.81 | 5(11.98) | 0.70 | 0.26–1.86 | 23(20.05) | 1.36 | 0.72–2.58 | 31(20.88) | 1.01 | 0.56–1.84 |
| • Secondary | 10(37.03) | 1.42 | 0.26–7.95 | 7(16.45) | 1.70 | 0.94–3.06 | 24(21.48) | 0.83 | 0.39–1.73 | 40(26.73) | 0.69 | 0.41–1.16 |
| • Higher | 6(22.22) | 0.07 | 0.01–0.88 | 2(4.45) | 2.54 | 0.21–3.72 | 4(3.95) | 1.43 | 0.48–4.27 | 12(7.91) | 0.51 | 0.21–1.22 |
| **Religion** | $X^2 = 4.47$ | | | $X^2 = 5.01^{\Psi\Psi}$ | | | $X^2 = 2.05$ | | | $X^2 = 1.88$ | | |
| • Muslim | 5(14.26) | 0.36 | 0.06–2.11 | 41(98.18) | 9.27 | 1.26–6.02 | 109(98.06) | 2.21 | 0.53–9.21 | 100(66.47) | 0.92 | 0.59–1.44 |
| • Others | 22(85.74) | Ref | | 1(1.82) | Ref | | 5(1.94) | Ref | | 50(33.53) | Ref | |
| **IPTp Doses Received** | $X^2 = 1.88$ | | | $X^2 = 0.44$ | | | $X^2 = 9.35^{\Psi}$ | | | $X^2 = 1.25$ | | |
| • 0–2 | 16(57.27) | Ref | | 16(36.98) | Ref | | 79(69.83) | Ref | | 78(52.13) | Ref | |
| • ≥3 | 11(42.73) | 0.58 | 0.25–1.30 | 26(63.02) | 1.39 | 0.59–3.24 | 34(30.17) | 0.48 | 0.28–0.83** | 72(47.87) | 0.83 | 0.55–1.44 |
| **Have Any Malaria prevention knowledge** | $X^2 = 0.65$ | | | $X^2 = 35.27^{\Psi\Psi\Psi}$ | | | $X^2 = 0.21$ | | | $X^2 = 0.45$ | | |
| • No | 5(15.08) | Ref | | 25(58.93) | Ref | | 13(12.11) | Ref | | 32(21.04) | Ref | |
| • Yes | 22(84.92) | 0.48 | 0.16–1.38 | 17(41.07) | 0.18 | 0.07–0.45*** | 100(87.89) | 0.81 | 0.34–1.93 | 118(78.96) | 0.97 | 0.61–1.55 |
| **Heard Any Malaria Massage in Media** | $X^2 = 6.26^{\Psi}$ | | | $X^2 = 1.65$ | | | $X^2 = 0.23$ | | | $X^2 = 5.32^{\Psi}$ | | |
| • No | 17(62.04) | Ref | | 31(75.19) | Ref | | 71(62.50) | | | 120(80.01) | Ref | |
| • Yes | 10(37.96) | 3.12 | 1.27–7.66* | 11(24.81) | 0.91 | 0.20–4.15 | 42(37.50) | 0.84 | 0.48–1.47 | 30(19.99) | 0.71 | 0.43–1.19 |
| **Has Access to Personal Transport** | $X^2 = 0.01$ | | | $X^2 = 0.40$ | | | $X^2 = 0.56$ | | | $X^2 = 11.19^{\Psi\Psi}$ | | |
| • No | 15(56.39) | Ref | | 21(50.05) | Ref | | 22(19.32) | Ref | | 56(36.92) | Ref | |

(*Continued*)

**Table 2.** (Continued)

| Characteristics | Central African Country | | | | | | West African Country | | | | | |
| | Cameroon | | | Guinea | | | Mali | | | Nigeria | | |
| | N (%) | AOR | 95% CI | N (%) | AOR | 95% CI | N (%) | AOR | 95% CI | N (%) | AOR | 95% CI |
|---|---|---|---|---|---|---|---|---|---|---|---|---|
| • Yes | 12(43.61) | 1.17 | 0.53–2.62 | 21(49.95) | 1.09 | 0.51–2.32 | 91(80.68) | 0.93 | 0.56–1.57 | 94(63.08) | 1.88 | 1.30–2.70** |

Denote: (N (%) = Number (percentage) of the women who received IPTp during antenatal visit categorized by factors, *P*-value indication

\* = *P*-value<0.05

\*\* = *P*-value<0.01

\*\*\* = *P*-value<0.001, Chi$^2$ significance

$^\Psi$ = *P*-value<0.05

$^{\Psi\Psi}$ = *P*-value<0.01

$^{\Psi\Psi\Psi}$ = *P*-value<0.001, AOR = adjusted odds ratio)

statistically significant in the multivariate analysis. No significant associations were observed for reproductive age groups in either the chi-squared tests or the multivariate models.

## Socio-economic factors

In Cameroon and Guinea, no significant associations were found in the chi-squared or multivariate analyses of the wealth index. In Mali, both the chi-squared ($X^2$ = 24.27, p<0.001) and multivariate analyses were significant, where poor women had higher IPTp coverage during ANC visits (AOR: 1.99, 95% CI: 1.04–3.81, p<0.05) compared to women in the middle wealth index. In Nigeria, significant associations were observed in the chi-squared test ($X^2$ = 8.41, p<0.05), but these associations were not significant in the multivariate analysis.

For access to personal transport, significant associations were found only in Nigeria in the chi-squared test ($X^2$ = 11.19, p<0.01). Women with access to personal transport had higher IPTp coverage (63.08%), and multivariate analysis showed they were significantly more likely to receive IPTp during ANC visits (AOR: 1.88, 95% CI: 1.30–2.70, p<0.01) compared to those without personal transport (36.92%). However, no significant associations were found in the other three countries—Cameroon, Guinea, and Mali—for access to personal transport in either the chi-squared or multivariate analyses.

## IPTp uptake factor

In Mali, a significant association was observed in the number of IPTp doses received during ANC visits both chi-squared test ($X^2$ = 9.35, p<0.01) and multivariate analysis. Specifically, 30.17% of women received 3 or more doses, while 69.83% received 0–2 doses. Women who received 3 or more doses were significantly less likely to receive during ANC visits (AOR: 0.48, 95% CI: 0.28–0.83, p<0.01) compared to those who received fewer doses. However, no significant associations were found in Cameroon, Guinea, and Nigeria.

## Cognitive factors

For malaria prevention knowledge, significant associations were found only in Guinea. The chi-squared test ($X^2$ = 35.27, p<0.001) and multivariate analysis showed that women with malaria prevention knowledge were significantly less likely to receive IPTp during ANC visits compared to those without such knowledge (AOR: 0.18, 95% CI: 0.07–0.45, p<0.001). No significant associations were found in Cameroon, Mali, or Nigeria in either the chi-squared or multivariate analyses.

For exposure to malaria-related messages in media, significant associations were observed in Cameroon ($X^2$ = 6.26, p<0.05). In Cameroon, women who heard malaria messages in the media were more likely to receive IPTp during ANC visits (AOR: 3.12, 95% CI: 1.27–7.66, p<0.05) compared to those who did not. In Nigeria, significant associations were found in the chi-squared test ($X^2$ = 5.32, p<0.05), but no significant associations were found in the multi-variate analysis. Additionally, no significant associations were found in Guinea or Mali for exposure to malaria messages in media in either the chi-squared or multivariate analyses.

## Discussion

This study provides important insights into the disparities in IPTp coverage during ANC visits across Cameroon, Guinea, Mali, and Nigeria, emphasizing the influence of several factors. While IPTp coverage was generally high, significant variations in uptake across different population groups highlight critical areas for policy and intervention focus. The substantial differences in IPTp coverage between urban and rural populations, particularly in Guinea and Mali, point to persistent healthcare access challenges in rural areas. Despite a higher percentage of rural women attending ANC in Guinea, they were significantly less likely to receive adequate IPTp doses. This reflects deeper systemic issues, such as limited healthcare infrastructure, staffing shortages, and possible cultural barriers in rural regions [29]. The findings emphasize the need for targeted interventions in rural areas, where possible solutions include mobile health services, community-based healthcare workers, and strategies to improve healthcare delivery infrastructure.

In contrast, Nigeria exhibited a relatively lower overall IPTp coverage, with access to personal transport emerging as a key factor for improving uptake. This underscores the logistical challenges that women in Nigeria face when accessing healthcare services [30]. Policies focused on enhancing transportation networks or providing transport subsidies could play a crucial role in improving the country's IPTp uptake and overall maternal healthcare.

The observation that poor women in Mali had higher IPTp coverage compared to their middle-income counterparts provides an unexpected but critical insight. This may be due to the successful targeting of poorer populations through government or NGO interventions [31]. These results prompt further research into understanding how socio-economic factors influence healthcare-seeking behavior, particularly among those in the middle-income bracket who may not fully benefit from public health programs aimed at the most vulnerable populations.

One of the most striking findings was the variation in IPTp dosing across countries, with a significant portion of women receiving suboptimal doses (0–2 doses). In Mali, where this issue was most prominent, women who received 3 or more doses were less likely to receive IPTp during ANC visits, pointing to potential inefficiencies in ANC delivery. This underlines the importance of reinforcing ANC protocols and ensuring consistent dosing guidelines are followed [32]. Health facilities must prioritize adequate IPTp dosing during every ANC visit, and governments should explore mechanisms such as digital tracking systems or automated reminders to healthcare providers to improve compliance with dosing schedules.

The role of cognitive factors in IPTp uptake, especially in Guinea and Cameroon, highlights the complex relationship between knowledge and action. While women in Guinea had high malaria prevention knowledge, they were paradoxically less likely to receive IPTp. This suggests that knowledge alone may not be sufficient to drive behavioral change. Potential explanations include mistrust in healthcare systems or misconceptions about the benefits of IPTp. This finding calls for more nuanced health communication strategies that not only disseminate information but also address underlying attitudes, fears, and cultural beliefs about healthcare

and pregnancy [33]. Health messaging needs to be clearer, consistent, and tailored to address the specific concerns of the population.

Meanwhile, in Cameroon, exposure to malaria-related messages through media was associated with higher IPTp uptake, underscoring the power of well-targeted health communication campaigns. However, despite high media exposure rates, there remain gaps in reaching all women, particularly those in rural or isolated communities. Expanding the reach of these campaigns and exploring alternative platforms like community radio or local leaders could enhance the effectiveness of malaria prevention strategies [34]. The findings of this study have significant implications for national health policies and international health programs focused on malaria prevention. Governments and health organizations must adopt a multifaceted approach to improve IPTp uptake, particularly in rural and underserved areas.

## Limitations

Despite providing valuable insights, the foremost limitation of this study is that the use of self-reported data to determine the receipt of IPTp during ANC visits introduces potential biases since responses can be subjective. Furthermore, the study's temporal scope is limited as the database was selected after 2020, which might only cover some insights before this period. However, it was necessary to omit countries with insufficient data or those that need to meet the criteria for inclusion, which may impact the overall generalizability of the findings. Another major limitation is that three countries are represented as a subset of West Africa and one country represents Central Africa, highlighting the constraints imposed by relying on this data as this is a major limitation to using secondary data as for this reason regional analysis is not suitable for such. This selection may need to capture the nuances in the broader African region. A more comprehensive understanding of influencing factors could be gained by considering other healthcare contexts. Additionally, by not exploring the cultural backgrounds of the participants, this study misses out on insights regarding patterns of receiving IPTp during pregnancy, as cultural nuances may play a significant role. These limitations also indicate areas for future research to explore to overcome constraints and enhance the understanding of the topic.

## Conclusions

This study identifies key disparities in IPTp coverage across Cameroon, Guinea, Mali, and Nigeria. Cameroon had the highest IPTp coverage, while Nigeria had the lowest. Rural women in Guinea and Mali were less likely to receive IPTp compared to urban women, possible healthcare access challenges in rural areas. In Nigeria, access to personal transport significantly improved IPTp uptake. In Mali, poorer women had higher IPTp coverage than middle-income women, suggesting targeted interventions may be reaching vulnerable populations. Suboptimal IPTp dosing was a concern, particularly in Mali, where many women received fewer than the recommended doses. Additionally, high malaria prevention knowledge in Guinea may not translate to higher IPTp uptake, indicating the need for more effective health communication strategies. These findings highlight the importance of improving healthcare infrastructure, enhancing transport access, and refining communication efforts to ensure better IPTp coverage in these regions.

## Acknowledgments

We express our gratitude to the Demographics and Health Survey (DHS) for granting us access to their survey dataset.

## Author Contributions

**Conceptualization:** M. D. Nahid Hassan Nishan.

**Data curation:** M. D. Nahid Hassan Nishan.

**Formal analysis:** M. D. Nahid Hassan Nishan.

**Methodology:** M. D. Nahid Hassan Nishan, Khadiza Akter.

**Visualization:** M. D. Nahid Hassan Nishan.

**Writing – original draft:** M. D. Nahid Hassan Nishan, Khadiza Akter.

**Writing – review & editing:** M. D. Nahid Hassan Nishan, Khadiza Akter.

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
