## [Editor Report · Decision Letter 0]

6 Sep 2024

PONE-D-24-37810Determinants of Intermittent Preventive Treatment in Pregnancy (IPTp) Coverage Among Pregnant Women During Antenatal Care Visits in Central and West AfricaPLOS ONE

Dear Dr. NISHAN,

Thank you for submitting your manuscript to PLOS ONE. After careful consideration, we feel that it has merit but does not fully meet PLOS ONE’s publication criteria number 3 as it currently stands. Therefore, we invite you to submit a revised version before it can be sent to external reviewers. Do address possible potential confounders and effect modifiers in the methods section which have to be accounted for in the regression analysis. If these cannot be reported, please provide an explanation. 

We look forward to receiving your revised manuscript.

Kind regards,

Adaoha Pearl Agu, MBBS, MSc, FMCPH

Academic Editor

PLOS ONE
---

## [Author Response · Author response to Decision Letter 0]

6 Sep 2024

Thanks a lot for your valuable feedback. The statistical section is updated clearly now.

---

## [Decision Letter · Decision Letter 1]

27 Sep 2024

PONE-D-24-37810R1Determinants of Intermittent Preventive Treatment in Pregnancy (IPTp) Coverage Among Pregnant Women During Antenatal Care Visits in Central and West AfricaPLOS ONE

Dear Dr. NISHAN,

Thank you for submitting your manuscript to PLOS ONE. After careful consideration, we feel that it has merit but does not fully meet PLOS ONE’s publication criteria as it currently stands. Therefore, we invite you to submit a revised version of the manuscript that addresses the points raised during the review process. Please address the issues raised by both reviewers. Publication criteria Number 3 with regards to the Materials and Methods and statistical analyses has not been adequately addressed. For example establishing whether the women had adequate as opposed to sub-optimal uptake of SP in addressing coverage is expected in the analysis and an explanation of why that was not/ cannot be done is necessary. Your current title suggests that a regional analysis should be done.

Please submit your revised manuscript by Nov 11 2024 11:59PM. If you will need more time than this to complete your revisions, please reply to this message or contact the journal office at plosone@plos.org. Please include the following items when submitting your revised manuscript:A rebuttal letter that responds to each point raised by the academic editor and reviewer(s). You should upload this letter as a separate file labeled 'Response to Reviewers'.A marked-up copy of your manuscript that highlights changes made to the original version. You should upload this as a separate file labeled 'Revised Manuscript with Track Changes'.An unmarked version of your revised paper without tracked changes. You should upload this as a separate file labeled 'Manuscript'.

We look forward to receiving your revised manuscript.

Kind regards,

Adaoha Pearl Agu, MBBS, MSc, FMCPH

Academic Editor

PLOS ONE

Reviewers' comments:

Reviewer's Responses to Questions

**Comments to the Author**

1. If the authors have adequately addressed your comments raised in a previous round of review and you feel that this manuscript is now acceptable for publication, you may indicate that here to bypass the “Comments to the Author” section, enter your conflict of interest statement in the “Confidential to Editor” section, and submit your "Accept" recommendation.

Reviewer #1: (No Response)

Reviewer #2: (No Response)

2. Is the manuscript technically sound, and do the data support the conclusions?

Reviewer #1: No

Reviewer #2: Partly

3. Has the statistical analysis been performed appropriately and rigorously? 

Reviewer #1: No

Reviewer #2: No

4. Have the authors made all data underlying the findings in their manuscript fully available?

Reviewer #1: No

Reviewer #2: Yes

5. Is the manuscript presented in an intelligible fashion and written in standard English?

Reviewer #1: No

Reviewer #2: Yes

6. Review Comments to the Author

Reviewer #1: Congratulations to the authors on this piece of valuable and timely manuscript. Malaria is a major killer disease in SSA. Sincere and dedicated efforts of researchers should be encouraged by providing constructive and impressive comments on their works, I feel, as this can contribute to a healthy research environment. In addition, thanks to this practice, researchers are inspired to work harder and harder.

However, I have major concerns with the quality of the paper, from the abstract, background, methods, analysis and results and discussion. The authors will have a great work to do to improve the quality of the paper. See my comments below:

1. The title of the paper is misleading. Since the authors analyzed data for only four countries, it should be stated as such: Cameroon, Guinea, Mali, and Nigeria. It is misleading to use only Cameroon to connote Central Africa, or three countries only to connote West Africa. This is unacceptable.

2. The title is redundant. IPTp is used during pregnancy, therefore, the title should be revise to something like; “Coverage and determinants of intermittent preventive treatment in pregnancy (IPTp) in Cameroon, Guinea, Mali, and Nigeria“.

3. Be consistent on whether to use “coverage” or “prevalence”.

4. “We found a higher prevalence of receiving IPTp during ANC visits, with Cameroon taking the lead at 98.59%”, please, state the prevalence for the other countries. Also, explain in the methods section clearly why you did not use the variable name ML1 – “Number of times took fansidar during pregnancy”, so you are able to establish whether women had adequate or sub-optimal SP uptake during pregnancy by categorizing to 0, 1-2 ≥3 or dichotomizing 0-2 vs. ≥3.

5. “We found a higher prevalence of receiving IPTp during ANC visits, with Cameroon taking the lead at 98.59%. Rural areas have a lower likelihood of receiving antimalarial treatment in Mali (AOR; 0.14) and Guinea (AOR; 0.16). Approximately 42.21% of lower-income individuals from Nigeria received IPTp during ANC visits. The study found that in Nigeria, personal transportation significantly increased the likelihood of receiving IPTp during ANC visits, highlighting the role of accessibility in healthcare utilization”. I found the results section of the abstract disconnected and poorly written.

6. “Rural areas in Mali and Guinea faced challenges due to limited healthcare infrastructure”, was healthcare infrastructure part of the data that you examined? How did you arrive at this conclusion? Neither of the authors are from any of these countries. Therefore, I suppose they are making wrong assertions.

7. “Interestingly, lower-income women in Mali were more likely to receive IPTp compared to their middle-income counterparts”, how did you measure “income”? To the best of my knowledge, MIS does not collect data on respondents’ or households’ income. Therefore, it is misleading for the authors to use “income” instead of the proxy variable used to measure household wealth.

8. It is clear that the authors do not have contextual information about the countries they studied. For example, Nigeria currently has the highest number of malaria cases globally, highest transmission rate, and deaths due to malaria. Nowhere in the introduction was the problem statement clearly stated. The entire introduction is poorly written without a good systematic analysis by the authors.

9. “The World Health Organization (WHO) now recommends three key malaria prevention strategies: seasonal malaria chemoprevention (SMC), perennial malaria chemoprevention (PMC, previously known as intermittent preventive treatment in infants, or IPTi)”, this is completely irrelevant.

10. “Despite these findings, it seems that there is a lack of comprehensive studies that identify factors that are contributing to the West and Central African region, which influences pregnant women receiving antimalarial medication during their ANC visits”, this justification for the study is completely untrue. There are several reasons you may want to conduct such study including to corroborate the findings of previous studies, maybe you are using a unique analytical method, or you want to refer to the advantage of pooling data from multiple countries in a single study. There are several of such rationale you can present.

11. “According to Wikipedia, there present 16 countries in West Africa and 9 countries in Central Africa”, so you are using only four countries out of 25?? Yet, you are using a title to mislead. Explain very clearly why you selected only four countries out of the 25.

12. “Study design & participants”, the entire sub-section is currently misleading. In the method section, you state how you extracted your study participants and the how you determine the study design.

13. Please, explain very clearly how you handle the missing data. What statistical method(s) did you use.

14. It is unclear how the authors measured IPTp in the dataset. What data element did you use? How did MIS measure the variable? This must be stated very clearly in the methods section. For example, MIS used variable M49 " During pregnancy took: SP/fansidar for malaria" with a sample size of 5497 women’s responses. How did you arrive at 11,120 sample size from the four countries? The sample size for each country must be clearly presented in the methods section.

15. “Outcome variable”, as stated earlier, this section should be written very clearly – including how the variable(s) was collected or named by MIS and how you manipulated the variable(s).

16. “Independent variables”, provide the framework(s) used to select the explanatory variables. This should include if you conducted a systematic analysis for variable selection.

17. “Statistical analysis”, this section still requires some revision.

18. It is difficult for me to relate with your results. They are so disconnected and shallow. You may seek the help of a statistician to re-analyze rigorously. As a biostatistician myself, I find these results grossly inadequate.

19. The discussion and conclusion sections still need a lot of work.

20. I find literature consulted inadequate. There are several papers from your study countries on this topic. You may do well to collaborate with a co-author from the region who can support with background information and contextual arguments in the discussion.

Reviewer #2: It is not clear what criteria was used to select "four countries were determined to be suitable." it appears that effort was made to find the most recent MIS (from 2021), but were there any other factors aside from being in West & Central Africa. One assumes that this region was selected because of its level of malaria transmission, but that is not stated as one would expect. Also there is no regional level analysis. As noted below this appears to be four individual studies joined together.

We see that "The outcome variable in this study focused on whether pregnant women received IPTp during their ANC visits," and this is dichotomized into Yes and No. The abstract unfortunately makes it sound like only the "Yes" people were studied. Please correct this: "The analysis focused on 11,120 women who received IPTp during their ANC visits."

Also in the abstract we see, "We found a higher prevalence of receiving IPTp during ANC visits..." One wonders "higher than what?" please re-phrase this to make sense. By the way, the prevalence of "YES" is high, but that comes from the fact that it is common for women to receive the first dose of IPTp but more problematic to get them to return for the 2nd, 3rd, 4th... and tis is where we really need information to improve programming and outreach!

The authors respond to the question of ethics by saying, "The study used a secondary Demographic and Health Survey (DHS) dataset. Hence, no ethical approval was required." The correct answer which should be included in the methods section, would be that DHS procedures include seeking consent. State briefly that this was done and how by the DHS program and the relevant Ministry.

Back to the issue of IPT Yes/No, it would therefore appear that the issue of number of doses received is not reported nor analyzed. Because of the nature of SP effectiveness over the years, WHO recommended that IPTp be given monthly from the very beginning of the second trimester, and at a minimum of 3 times during the pregnancy. One does may have some benefit, but at least 3 is the ideal. By not reporting and anlayzing this factor, the study misses an important health indicator.

the statement that "The decision to group women who received IPTp outside of ANC visits with those who did not receive IPTp at all stems from the study's focus on evaluating the effectiveness of ANC visits in delivering IPTp" makes sense as long as the authors consistently say that the dependent variable is receiving IPTp WITHIN ANC.

It would be worth reporting the DHS/MIS methodology that determines receipt of IPTp - use of cards, memory, etc. as well as describing what are those non-ANC sources of IPTp as any improvements in the programs on the ground depend on knowing more about what women do to see IPTp.

As an editorial concern, the authors often have one paragraph of text under a heading in the paper, and the narrative on variables is a case in point. Technically a header is justified only if it contains more than one paragraph. In this case one could have a header called "Definition of Study Variables" and put all information under just that one header. By the way, the paragraph for independent variables is large and needs to be broken into at least 2 paragraphs.

The results section is a bit confusing in that it almost appears that this is in fact 4 separate studies/four countries, pulled together under one set of results. This may arise because of the dependency on chi-squared. This leaves gaps and redundancies. For example we start demographic factors comparing "Guinea and Mali showed that those who lived in urban areas". Either this is a study that combines 4 recent MIS/DHS from a specific region, or 4 individual studies grouped into one article. One assumes that there was a reason for selecting 4 countries in one region - so what can we learn about the region and then about the individual countries?

Why noty at least take it variable by variable. For example, "When examining rural versus urban Residence, the MIS finding show that Mali ..., Guinea ..., Nigeria ..., and Cameroon..." Then do the same for Religion, Education, and the other independent variables.

If there were "a lower number of observations" in a particular category of a variable - regroup the categories.

Malaria knowledge and awareness are NOT socio-demographic variables. These are Cognitive/Perception factors.

In the Tables, make it easier to read by indenting the categories under each variable as for example -

Reproductive Age Group

*Early Reproductive Period

*Mid-Reproductive Period

*Late Reproductive Period

Much of the Discussion is a repetition or summary of the results. At this point we want to learn about the interpretations/meanings and the implications of the results.

Without addressing the issue of IPTp dosing, I really cannot say much more about the discussion.

The Conclusion states that "This study highlights significant disparities in IPTp..." without ,entioning those and what needs to be done specifically about each. We also see that, "limited healthcare infrastructure and shortages of healthcare providers" is a concern, but this was not one of the variables under study and outside the scope of what one would discuss and conclude.

Overall the MIS/DHS is always a righ source of data to help improve health and malaria programs. The authors have a unique opportunity to look at a region, not just individual countries. Hopefully they will reanalyze and make improvements so we can all learn from these studies.

7. PLOS authors have the option to publish the peer review history of their article (what does this mean?). If published, this will include your full peer review and any attached files.

Reviewer #1: No

Reviewer #2: No

---

## [Author Response · Author response to Decision Letter 1]

9 Oct 2024

Reviewer #1: Thank you very much for your thoughtful and constructive feedback on our manuscript. We sincerely appreciate your encouragement and detailed comments, which will help us improve the quality of our work. We are committed to addressing each of your concerns thoroughly to strengthen the paper as much as possible. Below are the responses, we made changes to the paper:

Congratulations to the authors on this piece of valuable and timely manuscript. Malaria is a major killer disease in SSA. The sincere and dedicated efforts of researchers should be encouraged by providing constructive and impressive comments on their works, I feel, as this can contribute to a healthy research environment. In addition, thanks to this practice, researchers are inspired to work harder and harder. However, I have major concerns with the quality of the paper, from the abstract, background, methods, analysis, results, and discussion. The authors will have a great work to do to improve the quality of the paper. See my comments below: 

Comment 1: The title of the paper is misleading. Since the authors analyzed data for only four countries, it should be stated as such: Cameroon, Guinea, Mali, and Nigeria. It is misleading to use only Cameroon to connote Central Africa, or three countries only to connote West Africa. This is unacceptable.

Response 1: Thank you for your valuable feedback. We have carefully revised the title.

Comment 2: The title is redundant. IPTp is used during pregnancy, therefore, the title should be revised to something like; “Coverage and determinants of intermittent preventive treatment in pregnancy (IPTp) in Cameroon, Guinea, Mali, and Nigeria “. 

Response 2: Thank you for your thoughtful suggestion. We agree and have revised the title accordingly.

Comment 3: Be consistent on whether to use “coverage” or “prevalence”. 

Response 3: Thank you for pointing that out. We have reviewed the manuscript and ensured consistent use of the term "coverage" throughout, as it is more appropriate.

Comment 4: “We found a higher prevalence of receiving IPTp during ANC visits, with Cameroon taking the lead at 98.59%”, please, state the prevalence for the other countries. Also, explain in the methods section clearly why you did not use the variable name ML1 – “Number of times took Fansidar during pregnancy”, so you are able to establish whether women had adequate or sub-optimal SP uptake during pregnancy by categorizing to 0, 1-2 ≥3 or dichotomizing 0-2 vs. ≥3.

Response 4: Thank you for your valuable suggestion. We have revised the abstract to include the prevalence for all countries as requested. Additionally, we have carefully considered the reviewers' comments regarding IPTp dose and have incorporated the variable "Number of times took Fansidar/Sp during pregnancy" (ML1) into our study. Following your recommendation, we have categorized the doses as dichotomous (0-2 vs. ≥3) to establish whether women had adequate or sub-optimal SP uptake during pregnancy, and the entire manuscript has been revised accordingly. Thank you again for your insightful feedback.

Comment 5: “We found a higher prevalence of receiving IPTp during ANC visits, with Cameroon taking the lead at 98.59%. Rural areas have a lower likelihood of receiving antimalarial treatment in Mali (AOR; 0.14) and Guinea (AOR; 0.16). Approximately 42.21% of lower-income individuals from Nigeria received IPTp during ANC visits. The study found that in Nigeria, personal transportation significantly increased the likelihood of receiving IPTp during ANC visits, highlighting the role of accessibility in healthcare utilization”. I found the results section of the abstract disconnected and poorly written.

Response 5: We have revised the result section of the abstract. Thank you.

Comment 6: “Rural areas in Mali and Guinea faced challenges due to limited healthcare infrastructure”, was healthcare infrastructure part of the data that you examined? How did you arrive at this conclusion? Neither of the authors are from any of these countries. Therefore, I suppose they are making wrong assertions. 

Response 6: Thanks for addressing this, We removed this line and revised the conclusion section. Thank you.

Comment 7: “Interestingly, lower-income women in Mali were more likely to receive IPTp compared to their middle-income counterparts”, how did you measure “income”? To the best of my knowledge, MIS does not collect data on respondents’ or households’ income. Therefore, it is misleading for the authors to use “income” instead of the proxy variable used to measure household wealth.

Response 7: Thank you for your valuable observation. We have removed the term "income" from the abstract and replaced it with the appropriate term "wealth index" throughout the manuscript, categorizing it into poor, middle, and rich groups. The detailed labeling method is provided in the Definitions of Study Variables section. We appreciate your helpful feedback.

Comment 8: It is clear that the authors do not have contextual information about the countries they studied. For example, Nigeria currently has the highest number of malaria cases globally, the highest transmission rate, and deaths due to malaria. Nowhere in the introduction was the problem statement clearly stated. The entire introduction is poorly written without a good systematic analysis by the authors. 

Response 8: We revised the entire introduction to resolve this issue. Thank you for your valuable feedback.

Comment 9: “The World Health Organization (WHO) now recommends three key malaria prevention strategies: seasonal malaria chemoprevention (SMC), perennial malaria chemoprevention (PMC, previously known as intermittent preventive treatment in infants, or IPTi)”, this is completely irrelevant. 

Response 9: We have removed and revised the line. Thank you.

Comment 10: “Despite these findings, it seems that there is a lack of comprehensive studies that identify factors that are contributing to the West and Central African region, which influences pregnant women receiving antimalarial medication during their ANC visits”, this justification for the study is completely untrue. There are several reasons you may want to conduct such a study including to corroborate the findings of previous studies, maybe you are using a unique analytical method, or you want to refer to the advantage of pooling data from multiple countries in a single study. There are several of such rationales you can present. 

Response 10: We kept your suggestion and revised the introduction section. Thank you.

Comment 11: “According to Wikipedia, there present 16 countries in West Africa and 9 countries in Central Africa”, so you are using only four countries out of 25?? Yet, you are using a title to mislead. Explain very clearly why you selected only four countries out of the 25.

Response 11: Thank you for commenting. Due to the limited availability of complete and relevant data across all 25 countries, only four countries with suitable 2021 MIS datasets were found to be appropriate for analysis. The study description has been revised to provide a detailed explanation.

Comment 12: “Study design & participants”, the entire sub-section is currently misleading. In the method section, you state how you extracted your study participants and the how you determined the study design. 

Response 12: Thank you for your comment. The entire Study Design & Participants section has been revised and written in detail to clearly explain how the study participants were extracted and how the study design was determined.

Comment 13: Please, explain very clearly how you handle the missing data. What statistical method(s) did you use? 

Response 13: Thank you. The handling of missing data, along with the statistical method used (listwise deletion), has been added and clearly explained in the revised Study Design & Participants section.

Comment 14: It is unclear how the authors measured IPTp in the dataset. What data element did you use? How did MIS measure the variable? This must be stated very clearly in the methods section. For example, MIS used variable M49 " During pregnancy took: SP/Fansidar for malaria" with a sample size of 5497 women’s responses. How did you arrive at 11,120 sample size from the four countries? The sample size for each country must be clearly presented in the methods section.

Response 14: This has been cleared on the revised Study Design & Participants section. Thank you.

Comment 15: “Outcome variable”, as stated earlier, this section should be written very clearly – including how the variable(s) was collected or named by MIS and how you manipulated the variable(s).

Response 15: This has been revised and shown in the Definition of Study Variables section. Thank you.

Comment 16: “Independent variables”, provide the framework(s) used to select the explanatory variables. This should include if you conducted a systematic analysis for variable selection.

Response 16: This also has been revised and shown in the Definition of Study Variables section. Thank you.

Comment 17: “Statistical analysis”, this section still requires some revision.

Response 17: We have revised the statistical analysis section to make it clearer and more precise, ensuring all methods are fully explained. Thank you for your suggestion

Comment 18: It is difficult for me to relate with your results. They are so disconnected and shallow. You may seek the help of a statistician to re-analyze rigorously. As a biostatistician myself, I find these results grossly inadequate.

Response 18: The result section is revised again including the data analysis. Thank you for your feedback.

Comment 19: The discussion and conclusion sections still need a lot of work.

Response 19: The entire discussion and conclusion section was revised thoroughly. Thank you for your comment.

Comment 20: I find the literature consulted inadequate. There are several papers from your study countries on this topic. You may do well to collaborate with a co-author from the region who can support you with background information and contextual arguments in the discussion.

Response 20: We revised both the background and discussion sections. Thank you.

Reviewer #2: Thank you for your valuable feedback and insightful comments. We greatly appreciate your input and we carefully address your suggestions to enhance the quality of our manuscript.

Comment 1: It is not clear what criteria were used to select "four countries were determined to be suitable." it appears that an effort was made to find the most recent MIS (from 2021), but were there any other factors aside from being in West & Central Africa. One assumes that this region was selected because of its level of malaria transmission, but that is not stated as one would expect. Also, there is no regional-level analysis. As noted below this appears to be four individual studies joined together. 

Response 1: Thank you for your comment. A detailed explanation of the selection process for the four countries is provided in the revised study description section. A regional-level analysis was not conducted because data for all countries in the region were not available, and attempting such an analysis with only four countries could lead to ambiguous and misleading conclusions. This is also a limitation of using secondary datasets. Therefore, the study presents findings for individual countries. We have also updated the title based on Reviewer 1's suggestion.

Comment 2: We see that "The outcome variable in this study focused on whether pregnant women received IPTp during their ANC visits," and this is dichotomized into Yes and No. The abstract unfortunately makes it sound like only the "Yes" people were studied. Please correct this: "The analysis focused on 11,120 women who received IPTp during their ANC visits."

Response 2: The abstract method section is revised. Thank you.

Comment 3: Also, in the abstract we see, "We found a higher prevalence of receiving IPTp during ANC visits..." One wonders "Higher than what?" please re-phrase this to make sense. By the way, the prevalence of "YES" is high, but that comes from the fact that it is common for women to receive the first dose of IPTp but more problematic to get them to return for the 2nd, 3rd, 4th... and this is where we really need information to improve programming and outreach!

Response 3: Thank you for addressing this. We revised the result section of the abstracts. Thanks for the feedback.

Comment 4: The authors respond to the question of ethics by saying, "The study used a secondary Demographic and Health Survey (DHS) dataset. Hence, no ethical approval was required." The correct answer which should be included in the methods section, would be that DHS procedures include seeking consent. State briefly that this was done and how by the DHS program and the relevant Ministry. 

Response 4: Thank you for this. An ethical approval section is added to the revised methodology. Thank you.

Comment 5: Back to the issue of IPT Yes/No, it would therefore appear that the issue of number of doses received is not reported nor analyzed. Because of the nature of SP effectiveness over the years, WHO recommended that IPTp be given monthly from the very beginning of the second trimester, and at a minimum of 3 times during the pregnancy. One does may have some benefits, but at least 3 is the ideal. By not reporting and analyzing this factor, the study misses an important health indicator. 

Response 5: Thank you for pointing out this significant indicator. We have now included the IPTp doses received as a variable in our study and have analyzed it by categorizing the doses as 0-2 vs. ≥3, in line with the reviewer's recommendation.

Comment 6: The statement that "The decision to group women who received IPTp outside of ANC visits with those who did not receive IPTp at all stems from the study's focus on evaluating the effectiveness of ANC visits in delivering IPTp" makes sense as long as the authors consistently say that the dependent variable is receiving IPTp WITHIN ANC. 

Response 6: We revised the entire section and removed this line. Thank you.

Comment 7: It would be worth reporting the DHS/MIS methodology that determines receipt of IPTp - use of cards, memory, etc. as well as describing what are those non-ANC sources of IPTp as any improvements in the programs on the ground depend on knowing more about what women do to see IPTp.

Response 7: Thanks for this. We revised the outcome variable section and added it to the Definition of Study Variables section. 

Comment 8: As an editorial concern, the authors often have one paragraph of text under a heading in the paper, and the narrative on variables is a case in point. Technically a header is justified only if it contains more than one paragraph. In this case, one could have a header called "Definition of Study Variables" and put all information under just that one header. By the way, the paragraph for independent variables is large and needs to be broken into at least 2 paragraphs. 

Response 8: Thanks for pointing out this. We kept your suggestion and revised it accordingly.

Comment 9: The results section is a bit confusing in that it almost appears that this is in fact 4 separate studies/four countries, pulled together under one set of results. This may arise because of the dependency on chi-squared. This leaves gaps and redundancies. For example, we start with demographic factors comparing "Guinea and Mali showed that those who lived in urban areas". Either this is a study that combines 4 recent MIS/DHS from a specific region, or 4 individual studies grouped into one article. One assumes that there was a reason for selecting 4 countries in one region - so what can we learn about the region and then about the individual countries?

Response 9: Thank you for this valuable feedback. To clarify, this study involves the analysis of four separate countries, and a detailed methodological approach is provided in the study description section, explaining the rationale for selecting these countries. Initially, we aimed to conduct a region-wide analysis, but due to data limitations, which are noted in the limitation section, we shifted our focus to country-specific analysis. Consequently, we revised the title to reflect this change. Whi

---

## [Editor Report · Decision Letter 2]

18 Oct 2024

Coverage and Determinants of Intermittent Preventive Treatment in Pregnancy (IPTp) in Cameroon, Guinea, Mali, and Nigeria

PONE-D-24-37810R2

Dear Dr. MD NAHID HASSAN NISHAN,

We’re pleased to inform you that your manuscript has been judged scientifically suitable for publication and will be formally accepted for publication once it meets all outstanding technical requirements.

Kind regards,

Adaoha Pearl Agu, MBBS, MSc, FMCPH

Academic Editor

PLOS ONE
---

## [Editor Report · Acceptance letter]

22 Oct 2024

PONE-D-24-37810R2 

PLOS ONE

Dear Dr. NISHAN, 

I'm pleased to inform you that your manuscript has been deemed suitable for publication in PLOS ONE. Congratulations! Your manuscript is now being handed over to our production team.

Kind regards, 

on behalf of

Dr. Adaoha Pearl Agu 

Academic Editor

PLOS ONE